# Ionizing Radiation and Complex DNA Damage: From Prediction to Detection Challenges and Biological Significance

**DOI:** 10.3390/cancers11111789

**Published:** 2019-11-14

**Authors:** Ifigeneia V. Mavragani, Zacharenia Nikitaki, Spyridon A. Kalospyros, Alexandros G. Georgakilas

**Affiliations:** DNA Damage Laboratory, Department of Physics, School of Applied Mathematical and Physical Sciences, National Technical University of Athens (NTUA), 15780 Athens, Greece

**Keywords:** ionizing radiation, complex DNA damage, Monte Carlo simulations, detection of DNA damage, fluorescence microscopy, biological response, radiation biology

## Abstract

Biological responses to ionizing radiation (IR) have been studied for many years, generally showing the dependence of these responses on the quality of radiation, i.e., the radiation particle type and energy, types of DNA damage, dose and dose rate, type of cells, etc. There is accumulating evidence on the pivotal role of complex (clustered) DNA damage towards the determination of the final biological or even clinical outcome after exposure to IR. In this review, we provide literature evidence about the significant role of damage clustering and advancements that have been made through the years in its detection and prediction using Monte Carlo (MC) simulations. We conclude that in the future, emphasis should be given to a better understanding of the mechanistic links between the induction of complex DNA damage, its processing, and systemic effects at the organism level, like genomic instability and immune responses.

## 1. Introduction

Ionizing radiation (IR) is considered to be an environmental or anthropogenic genotoxic agent and has a varying quality depending on its type, i.e., electromagnetic or particles, etc. The most well-accepted characteristic quality parameter of IR is the linear energy transfer (LET) with values varying from ~1 keV/µm (low-LET: X, γ-rays) to higher (10 keV/μm for radiation therapy protons, or 10–100 keV/µm for carbon ions) and very high values (>100 keV/µm) for alpha-particles or heavy charged particles, like space radiation. The phenomenal paradox of alpha particles with higher LET than carbon ions, which are heavier, is attributed to the fact that the latter are accelerated in order to be used in therapy, while the LET of alpha particles refers to their production by spontaneous emission taking place during alpha decay. LET increases with the ion size (for the same energy) and decreases with increasing velocity-kinetic energy (of a given ion). As a charged particle (ion) traverses matter, it loses energy and thus its LET increases resulting in the maximum dose deposition in the so-called Bragg peak, or the clinically used spread-out Bragg peak (SOBP). Biological effects (RBE: relative biological effectiveness) are expected to maximize in this and distal end area, i.e., towards the end of the particle path [1,2]. These radiation physics and biology characteristics are considered the fundamental basis of current proton and carbon therapy advantages compared to commonly used X-ray-based treatments [3,4,5,6,7].

Besides LET, nowadays’ IR research focuses also on the particle trajectories themselves, since particles characterized by the same LET may induce similar biological effects along the same penetration length. Stochastic events of energy transfer, radial generation, and propagation of secondary free radicals occurring transversely to the trajectory are consistently studied using Monte Carlo (MC) simulations and hadronic calorimeters [8,9]. A particle trajectory is now considered as a “cylinder” of ionization events, the radius and the “content” of which affects the IR biological consequences. This practically means that if the same LET is to be attributed to cylinders of the same length but of different radius, the particle with the smaller ionization radius would trigger more ionizations per unit volume, causing more complex DNA lesions [8,10]. IR has a unique characteristic, which differentiates it from any other type of endogenous or exogenous (to the cell) stressors like replication stress or environmental chemicals, respectively, that is the complexity of damage induced in all cellular compartments (nucleus, cytoplasm, membranes, etc.).

The complex (clustered) nature of IR induced damage stems directly from the physical principles of IR interaction with matter. More specifically, IR has the ability to create ionization cascades upon its interaction with any material, including the usually highly hydrated biological materials (cells and tissues) [11]. This idea of the association of IR biological significance with ionization and damage clustering began as a speculation in the 1980s; since then, it has increasingly been supported by using primarily MC simulations [12,13,14,15,16,17,18]. Several years later, these theoretical suggestions were confirmed by solid experimental data, showing, for example, increased DNA fragmentation as the result of these dense energy depositions, increasing with LET and thus providing support for DNA damage clustering [19,20,21,22,23,24,25,26]. In other words, it gets more and more accepted that IR induces its major lethal or mutagenic effects not solely by the presence of simple DNA damage like single-strand breaks (SSBs), base lesions, and even a double-strand break (DSB), but primarily by the induction of clustered spatiotemporal damage to DNA. There is plenty of evidence pointing towards the importance of a radiation-induced critical ‘clustered lesion’, which consists of two or more types of DNA damage in a small DNA neighborhood of a few base pairs (bp), e.g., a DSB and a non-DSB oxidized guanine or a DSB and a SSB closely situated. The reader can refer to earlier and recent analytical reviews, which emphasize and back up this notion [16,27,28,29,30].

The increasing knowledge of the role of damage complexity elevates the biological significance of the various types of IR, from low (<0.1 Gy) to high doses normally used in clinics (2–3 Gy). In other words, it signifies the necessity to study in depth and improve detection methodologies for delineating the role of complex DNA damage in determining the outcome of biological and, most probably, clinical response of tumor and normal tissue [31,32]. In the next sections, we provide a detailed description, within the limitations of space, of the evolution and progress in complex DNA damage prediction, detection, and the evidence for its biological and clinical significance.

## 2. Radiobiological Modeling and Simulations: A Useful Tool for Prediction and Experimental Data Interpretation

In order to explain various radiobiological findings, several empirical deterministic models have been developed so far, contributing to the interpretation of the general trend of the experimental data. Although these models provide satisfactory qualitative results at relatively high doses, they cannot always contribute efficiently to corresponding outcomes at low doses [33]. The majority of these models are macroscopic, relating cellular time points to dose and other parameters associated with the irradiated cell, and irradiation conditions. The most characteristic example of these models is the linear quadratic cell survival model (LQ), which is widely used in conjunction with experiments [33,34]. Other models, such as the local effect model (LEM) [35] and the microdosimetric kinetic model (MKM) [36], take into account the LET and other quality characteristics of the irradiating particles and the tracks of the secondary particles in the calculation of the cell survival curves. Besides these models, the BIANCA model, in addition to cell survival, also predicts chromosome aberrations, which can be regarded as a strong indicator of normal tissue damage [3,37].

### 2.1. Monte Carlo Simulations for DNA Damage

MC methods represent a wide class of numerical computer simulation techniques that use statistical resampling to solve complex systems that are not easily tractable analytically and contain stochastic events. Generally, the MC method is a simulation technique in which the computer experiments are achieved by generating pseudorandom numbers repeatedly. Such simulations require large quantities of random numbers. If the probability law for the basic processes of the described phenomenon is known, these processes are generated on the computer in such a way as if they actually occur. An MC simulation is able to reliably reproduce the complex radiation transport stochastic phenomena without any major approximations or assumptions [38].

Focusing on the underlying biological mechanisms of cellular response to radiation, MC codes have been constructed to simulate damage induction at the DNA scale. Many theoretical studies, along with MC track structure calculations (MCTS), have contributed a lot to the understanding and the consolidation of the particularities between radiations of different qualities, the quantification of the dynamics of clustered DNA damage, and efforts to produce model-guided experiments. MCTS simulate the physical interactions creating “direct” damages to DNA, as well as simulations of the production and initial reactions of chemical species causing ‘‘indirect’’ damages. Thus, track structure codes are useful tools for the simulation of particle tracks in biological matter, providing a large experience in estimating the parameters of radiation effects relative to radiotherapy (RT) and radiation protection.

### 2.2. Modeling of Radiation Effects in Biological Material

In principle, the interaction of radiation with matter and living cells involves the transfer of energy from the particular radiation type to atoms and molecules of the irradiated matter. This occurs through various modes of atomic and nuclear interactions in the form of inelastic and elastic scatterings. The inelastic interactions are the events producing excitations and ionizations in matter, whilst the elastic scattering is a nuclear interaction causing scattering of the particles. The latter determines the shape of each track [39]. MCTS simulation is based either on the solution of analytical equations describing the transport of charged particles in the biological medium, or on a numerical solution by sampling the model of the particle interactions with the atoms and molecules of the medium. There are various ways of describing radiation transport and charged particle tracks of photons, electrons, neutrons, and ions: (a) 1D description of track, based on a single parameter of track (e.g., LET) and relevant parameters of radiation quality (e.g., RBE); (b) amorphous description of track based on average radial dose profiles around the particle trajectories; (c) condensed-history-method codes providing averages over discrete particle histories (they account for the cumulative effect of a large number of interactions along a small segment of the track and, being computationally efficient, they are preferred for the transport of high-energy electrons―i.e., for energy higher than 1 MeV); (d) the full 3D MCTS, which simulates the individual interactions between radiation and atoms of the matter and provides detailed distributions of elastic and inelastic interactions; and (e) MC simulation of charged particle track in condensed media and the evolution of free radicals produced in the biological matter with time (4D codes) [40,41].

In MCTS simulations, the stochastic nature of radiation action is completely taken into account using interaction cross sections, associated with interaction probabilities. More specifically, these simulations use the quantity ‘cross section’ (CS) to represent the probability of elastic and inelastic interactions of a charged particle with atoms of the irradiated matter. In this way, such a model of track structure consists of descriptions for these kinds of scattering in terms of different CSs, such as ‘total elastic’, ‘total inelastic’, ‘total excitation’, ‘total ionization’, ‘singly differential’ CS, ‘doubly differential’ CS, ‘charge-transfer’ CS, ‘multiple ionization’ CS, ‘dissociation’ CS, and ‘stopping power’ [42]. Water vapor is the medium in which the majority of experimental data has been collected, while relevant data of other materials (proteins, DNA, etc.) are practically very difficult to measure [43]. Therefore, CSs have been evaluated through various theoretical approaches. For ions and secondary electrons, energy depositions are mainly due to ionizations and excitations of target molecules. As ions slow down and approach the Bragg peak, interactions such as charge exchange and elastic scattering become increasingly significant. Energy depositions by these interactions affect the tracks of low- and intermediate-energy ions. Above all, charge exchange reactions modify the charge state of ions, leading to changes in the strength of subsequent interactions of these projectiles [44].

In theory, track-structure codes are used to simulate the physical interactions following the passage of the incident particle through biological matter for obtaining energy depositions and ionizations of the nanometer scale in precise models of the cells. After these interactions, the created radical species, together with ionized molecules, react in the physico-chemical stage, which is followed by transport of the chemical products in the same scale. During this stage, the latter react with each other and affect the components of the cell in their own way. The whole process terminates by determining the direct and indirect damages to the major target of the cell: the genetic molecule [45]. These damages, during the G0/G1 phase of the cell cycle, can produce chromosome aberrations due to incorrect rearrangement of chromosome fragments. Thus, some codes are extended to the level of providing chromosome aberration dose–response curves [3].

### 2.3. Radiation Transport and Track-Structure Codes for Radiation-Induced Damage

MCTS simulations nowadays offer the most sophisticated tool, in theory, for studying radiobiological interactions especially where experimental measurements are unachievable. These codes aim to achieve the simulation of sub-keV secondary electrons, which are associated with high radiobiological efficacy [46]. Currently, the degree of sophistication of these codes has been expanded further by simulating DNA damage and repair pathways. Clustered damage tries to correlate the different types of DNA damage frequencies of occurrence with the observed biological lesions. As a consequence, these codes are based on the classification of damage by energy deposition and by the number of ionizations and hydroxyl radicals in the genetic molecule. Up to now, several MC simulation codes can provide the nanoscale track structure of particles passing through biological matter, which is typically simulated as water but more recently also includes DNA nucleotide material. The most widely used among these codes, which are also accessible for public use, are presented in Table 1, along with an overview of their main advantages and disadvantages.

Another aspect of current research focuses on the use of a molecular dynamics approach such as in the MBN Explorer [67] to simulate the interactions between molecules. However, due to the fact that this research aims to increase the simulation detail, it inevitably leads to the complexity of simulations. Ergo, researchers often apply a multi-scale approach shifting to the nm-scale simulations of particle tracks and yielding clustered DNA damage of the irradiated cell. Such approaches that could be mentioned are Geant4/Geant4-DNA or the multiscale approach (MSA) [68]. Moreover, the possibility to manage parametrized volumes acquired from fluorescence microscopy in the Geant4 simulation toolkit at the sub-μm scale allows the assessment of the energy deposit and specific energy in various cell compartments. In this context, quite recently, the experimental approach was coupled to MC simulations to provide a way to selectively irradiate a single cell in a rapidly dividing multicellular model with a reproducible dose, thus allowing the MC dosimetry and calculation of the energy deposit in the nucleus and chromatin of the specific system [69].

Ionization of DNA generates direct damage to the genetic macromolecule, while that of the cytosol results in the formation of reactive species. The latter, such as hydroxyl radicals, hydrated electrons, and hydrogen atoms, diffuse to the nm-space around the ionization event, react inevitably with the DNA constituents, and generate lesions to them indirectly. Part of these lesions is localized on the strand(s) of the genetic molecule forming sites referred to as *clustered DNA lesions* or *multiply damaged sites*. This type of damage, due to its endogenous nature, is associated with error prone repair and may result in non-reversible cell damage, compared to sparsely distributed lesions. For example, a DSB consists of two SSBs, which are located on opposite strands within 10 bp of each other [10]. Clustered DNA lesions were first hypothesized to exist in the 1990s. Contemporary research has progressed to the point where these lesions have been characterized and measured in irradiated as well as in non-irradiated cell-targets. In 1988, John Ward put forward the idea of locally multiplied damaged sites and suggested that the difference in lethality of all the damaging agents was because of the spatial distribution of lesions and the cell inability to repair these clusters of damage [70,71].

A clustered DNA lesion can consist of single- and double-strand breaks, as well as base damage and abasic sites. These damages can be located on one strand (which includes tandem lesions) or on opposite strands (bistranded lesions). They include DSB clusters, non-DSB clusters, and tandem lesions (i.e., two damages situated immediately next to each other in the same strand of DNA). Double lesions consist of two damages that are situated on adjacent nucleosides or on one nucleoside. Another type of clustered lesion is the one that consists of an SSB and an adjacent interstrand crosslink with a cytosine on the opposite strand [72]. Bistranded clusters have been classified into two groups: Non-DSBs, which consist of base damage, AP sites, together with SSBs; and DSB clusters, which have oxidative damage situated near the DSB. The non-DSB clusters are specified into oxypurine (purine base damage), oxypyrimidine (pyrimidine base damage), and abasic site clusters (containing AP sites). It is of note that the higher the LET, the denser the ionization track and, thus, the greater the probability of complex lesions to be produced [73].

An increase in the complexity of the damage may consist of tandem or double lesions in opposition to damage in the opposing strand. MC codes predict that high LET radiation can generate clustered lesions consisting of 10–25 lesions over a 100–200 bp DNA region [74]. However, these sites of DNA damage should not be considered in isolation. Correlation of such lesions can result in complex gene mutations and complex chromosome rearrangements [75,76]. Track structure codes enable us to calculate the energy deposited at the nanometric scale, modeling particle tracks in gaseous media or liquid water. On the other hand, experiments are not capable of resolving the DNA damage at a single bp level. Therefore, track structure codes are the only method capable to account for the DNA damage at a single bp.

On the whole, all these codes frequently differ in their underlying assumptions, with different effectuations of particle transport and physics. Thus, simulated patterns of damage in the DNA must be combined with models that describe the mechanisms of DNA repair. There can also be serious differences among scientific communities in the way DNA damages are defined. Based on this fact, recently it has been proposed a new standard DNA damage (SDD) data format aiming to unify the interface between the simulation of damage induction in DNA and the biological modeling of DNA repair processes. It also introduces the effect of the environment of the cell as a flexible parameter in the whole procedure [77]. Studies have indicated that the predominant features of radiations can be usefully considered in terms of four classes of initial clustered damage of increasing severity: Class 1 damage includes a sparse initial physical damage with few tens of eV of energy deposited within ~2 nm and is predominantly SSBs. Class 2 damage is characterized by moderate clusters with an energy deposition of ~100 eV within ~2 nm of the DNA target and is mostly of DSBs (characteristic of low-LET radiation, being repairable). Class 3 damage comprises large clusters with ~400 eV of energy within 5–10 nm and is of more complex DSBs (characteristic of low-LET, also being repairable). Finally, class 4 damage includes very large clusters (~800 eVs are deposited within 5–10 nm of the target) and consists of complex types of DSBs [78].

### 2.4. Key Conclusions

Some basic conclusions about the simulation of clustered DNA damage by the use of MC codes are the following [42,79,80]: 

Substantial proportion of dose by low-LET radiation is deposited via low-energy electrons producing clustered DNA damage.

The majority of damage events are of simple type containing an SSB. Strand breaks of greater complexity appear at high frequencies.The complexity of damage increases with LET.The yield of both single- and double-strand breaks per gray and per Dalton is nearly constant over a wide range of LETs.For low-LET radiations, nearly 20% of the DSBs are of complex type. The proportion of this clustered damage increases with LET, reaching ~70% or higher for high-LET radiations.When base damage is also taken into account, the proportion of complex DSBs increases for all radiations, reaching more than 90% for the higher LETs.Experiments show a slower rate of repair of DSBs produced by high-LET radiations.

## 3. Detection of Complex DNA Damage

### 3.1. The In Situ Detection of Complex DNA Damage and the Colocalization Concept

Some of the first DNA damage detection techniques were based on electrophoresis or flow cytometry [81]. They use the collective signal from thousands of molecules and they can be performed in vitro, but not in situ. On the other hand, cytogenetics and in situ immunofluorescence (like γ-H2AX foci assay) provide the ability of damage detection at the site they occur. The in situ detection of complex DNA damage relies on the colocalization concept, additionally to all of the “DNA damage in situ detection” principles. Colocalization is, by definition, the presence of two or more fluorophores at the same physical location. In other words, colocalization can be described as the spatio-temporal overlap of two or more dyes in a multichannel image. The biological meaning is that two or more (macro-) molecules of different kinds, labeled with fluorophores, and exhibiting colocalized signal are connected to the same subcellular structure. This proximity―when not spontaneous―implies potential biological interaction between the labeled molecules.

The experimental orchestration of complex DNA damage detection in situ includes the following five modules, as they are also presented in Figure 1: The damaging factor (irradiation, chemical reagent, or genetically engineered clones);The visualization target (the damage itself or a participating repair enzyme);The labeling method (immuno-labeling or fluorescent labeling);The imaging apparatus (static or live cell imaging);The image analysis (static or tracking).

### 3.2. Damage Induction

Since the interest for clustered/complex DNA damage emerged from radiation biology, the most common damaging cause is radiation, and usually IR. Sophisticated irradiations―targeting a subcellular area―can be performed utilizing a laser microbeam [82,83], or even an ion microbeam [84,85]. In addition to the exposure of the biological material to irradiation, occasionally for comparison reasons, the latter is exposed to chemical agents [86] or is genetically engineered [87].

### 3.3. Damage Visualization: Direct or Indirect

The most common way to visualize (complex) DNA lesions is via labeling DNA repair enzymes. The predominant technique is immunofluorescence, where antibody technology is used to target the object to be detected. The most widely used enzyme using antibody technology is the phosphorylated histone H2AX, i.e., γH2AX, γ-H2AX, or gamma H2AX. There is also the option of labeling the damage itself. Suitably developed antibodies have the ability to bind onto specific DNA modifications, e.g., anti- 8-oxodG antibody for the detection of base lesions [88].

### 3.4. Labeling Techniques

#### 3.4.1. Immunolabeling—Fixed Cells

##### In Situ Immunofluorescence-Fluorophore Conjugated Antibodies

The predominant method for the in situ clustered DNA damage detection is immunofluorescence. Damaged cells, after repair for a specific period of time, are incubated with antibodies against the targets of interest; the antibodies used are conjugated with fluorophores. Clustered DNA damage genome regions are revealed when antibodies against DNA repair enzymes of different DDR (DNA damage response) pathways are used [23,25,89,90,91,92].

##### Transmission Electron Microscopy (TEM) Analysis—Nanoparticle Conjugated Antibodies

Colocalization analysis via TEM follows the same concept as in situ immunofluorescence assay. The telling difference between the two methods is the labeling molecule; here, antibodies are conjugated with nanoparticles. The exploration of colocalization events is achieved by the use of nanoparticles of dissimilar size [24,26,93].

##### Proximity Ligation Assay―In Situ PLA

Protein colocalization can also be visualized by the one-and-only fluorescent dye, unlike traditional colocalization techniques, which utilize one dye for every different target-enzyme. This assay is called proximity ligation assay (PLA) and as its name proclaims, fluorescent signal is produced only when the molecules of interest are found to be located in close proximity. Α fluorescent macromolecule is then synthesized, producing enough fluorescence to be visualized under a conventional fluorescence microscope. PLA assay has the ability to make the proximity of two single molecules visible [94,95,96].

#### 3.4.2. Live Cell Imaging

Fluorescence labeling of target proteins in live cells facilitates visualizing their intracellular distribution and interactions, as well as their time evolution (dynamics). Here, the main methods with regard to DNA repair are presented.

##### Encoding Fluorescence Labeled Proteins

The principle behind live cell imaging is to induce cells to express the desired proteins (here, DDR-associated ones) that in addition to their properties, are also fluorescent. To this end, a (set of) suitable plasmid(s) has to be transfected into the cells, carrying the modified gene(s) of the protein(s) of interest. As a result, and upon DNA damage, the cells produce the DDR proteins that conserve their initial properties, but they are also fluorescent, thus allowing their visualization via light microscopy. Such proteins are called recombinant fusion proteins labeled with fluorescent proteins (FP) or autofluorescent proteins (AFP) [88].

##### Fluorogenic Dyes

Fluorogenic or fluorigenic dyes are―as their name proclaims―not fluorescent themselves, but become fluorescent in a complex with the molecule of interest. They are based on antibody technology to identify the proper molecule and to form a fluorescent complex with it. However, the substrate that they are added to (i.e., cells) needs to be genetically engineered. If, for example, the target of interest is a protein, then a tagging gene needs to be added next to the gene sequence of the protein of interest. Thus, the target protein is expressed along with its tag. This tag is recognized by a fluorogen, and then labeling is achieved. Fluorigenic dyes of DNA repair enzymes are reported in several articles [97,98,99,100].

### 3.5. Imaging: Microscopy and Image Analysis

#### 3.5.1. Microscopy: TEM and Fluorescence Microscopy

TEM microscopy, although it offers an impressive resolving power down to a few nanometers, is not very popular for the study of DNA damage or repair, due to its “inherent” disadvantages; TEM requires the presence of high vacuum and a specimen slicing thinner than cell thickness, making the observation of live cells impossible. Moreover, specimen preparation protocols are quite demanding in terms of experimentalist’s technical skills. Another limiting factor is the TEM apparatus cost (~100k USD). On the other hand, fluorescence microscopy is suitable for the observation of both fixed and live cells; its resolving power and subsequently its cost vary among the different modules, which are shortly reviewed here. The reader may also refer to [101] for technical details in optical microscopy regarding live cell imaging.

##### Conventional (Widefield) Fluorescence Microscopy

Conventional microscopy, i.e., traditional microscopy (MS), is characterized by the illumination of the entire specimen simultaneously. It is also referred to as widefield MS (WFM), to be differentiated from confocal MS where the specimen is illuminated partially and sequentially. WFM was the first fluorescence MS system, and nowadays is the most widely used technique, as it has become cost efficient (beginning from a few k USD). WFM resolving power is theoretically diffraction limited (~200 nm axial and ~500 nm lateral), but in practice, it is much weaker.

##### Confocal Microscopy: Scanning Laser or Spinning Disk

In confocal microscopy (CFM), the illumination is accomplished by lasers. Since the lasers emit in one wavelength, no excitation filter is needed. Optical sectioning of the specimen is achieved by CFM. Scanning laser CFM was the first confocal adaptation. Later, spinning disk technology came up. In spinning disk, the single pinhole is replaced by two or more pinholes on the same aperture (light source), allowing more than one voxel to be simultaneously illuminated, thus accelerating the image acquiring process [82]. CFM offers better resolution than WFM, but is still diffraction limited; its cost begins from ~100k USD.

##### Super Resolution Microscopy (SRM): Beyond the Diffraction Limit

SRM relies on the invention of new materials, the metamaterials, with negative refracting index, i.e., negative permittivity and magnetic permeability. Lenses of this type of metamaterials are called superlenses. A superlens can propagate not only the far-field part of emitted light, but also the near-field (evanescence waves). Evanescence is the part of emitted signal that contains information about the object details that are smaller than the diffraction limit. The reader may refer to the review by Schermelleh et al. [102] for detailed information regarding SRM. Key SRM applications for DNA damage detection can be found in [103,104,105,106,107,108,109,110]. SRM resolution is ~100 nm axial and ~300 nm lateral, while it costs several hundreds of k USD.

#### 3.5.2. Image Analysis: Co-Localization Coefficients, Parameters, and Methods

Common colocalization approaches like *Pearson correlation coefficient* [111], *(Manders) Overlap coefficient, r* [112], *(Manders) Overlap coefficient, r^2^, k_1_ and k_2_* [112], *(Manders) M_1_ and M_2_ overlap coefficient* [112], Costes’ automatic threshold [113,114], *Van Steensel’s CCF*, or *Cytofluorogram creation*, correlate the pixel intensity between the fluorescence channels. Such approaches provide results with regard to a ‘given’ image, and not to a ‘given’ cell. However, the core of complex DNA damage analysis is the cell nucleus and especially the DSB-detection fluorescence focus. Some approaches, based on the DSB focus topology, are discussed here:

##### Defining an Extra Type of Foci

A quite simple method―to investigate the degree of colocalization in a given cell population―is to introduce an extra focus type―let us call it “colocalization focus”―in addition to DSB foci. Colocalization foci conserve all the detection characteristics of the given DSB focus type, along with an intensity threshold criterion for the second channel (i.e., the non-DSB foci channel when clustered DNA damage is to be detected). By determining the ratio:$$y=\frac{\mathrm{number}\;\mathrm{of}\;\mathrm{colocalization}\;\mathrm{foci}\;\mathrm{per}\;\mathrm{cell}}{\mathrm{number}\;\mathrm{of}\;\mathrm{DSB}\;\mathrm{foci}\;\mathrm{per}\;\mathrm{cell}}\;,$$
the colocalization degree is estimated. 

##### 
$R_{\mathrm{col}}$


A second quantitative colocalization parameter is the colocalization ratio $R_{\mathrm{col}}$, introduced by Martin et al. [115] and expressed as:$$R_{\mathrm{col}}=\frac{2I_{53\mathrm{BP}1}}{{Ι}_{\gamma H2\mathrm{AX}}\;+{\;I}_{53\mathrm{BP}1}}$$
where I_53BP1_ is the average per pixel intensity of the 53BP1 channel for a given γH2AX focus and ${Ι}_{\gamma H2\mathrm{AX}}$ is the analogous entity for the γH2AX channel. Although $R_{\mathrm{col}}$ was introduced to estimate the degree of colocalization between the two DSB repair proteins γH2AX and 53PB1, it could be also used for the clustered DNA damage detection, in terms of the detection of non-DSB repair enzymes in the vicinity of a DSB. Therefore, $R_{\mathrm{col}}$ could have the general formula of:$$R_{\mathrm{col}}=\frac{2I_{\mathrm{nonDSB}\;\mathrm{repair}\;\mathrm{enzyme}}}{{Ι}_{\mathrm{DSB}\;}+{\;I}_{\mathrm{nonDSB}\;\mathrm{repair}\;\mathrm{enzyme}}}$$
where $I_{\mathrm{nonDSB}\;\mathrm{repair}\;\mathrm{enzyme}}$ and ${Ι}_{\mathrm{DSB}\;}$ would be the averages per pixel intensity for the non-DSB and DSB repair enzyme channel, correspondingly, for a given DSB focus.

##### 
$P_{\mathrm{clc}}$


Another colocalization parameter is the $P_{clc}$
*parameter*, which was introduced by our group [116] and compares the intensity of the non-DSB protein fluorescence signal between the DSB foci area and the rest of cell nucleus; i.e., the cell nucleus area (volume) excluding DSB foci area (volume). $P_{\mathrm{clc}}$ is described by the expression:$$P_{\mathrm{clc}}=\frac{\mathrm{average}\;\mathrm{mean}\;\mathrm{Intensity}\;\mathrm{of}\;\mathrm{nonDSB}\;\mathrm{channel}\;\mathrm{over}\;\mathrm{DSB}\;\mathrm{foci}\;\mathrm{volume}}{\mathrm{average}\;\mathrm{mean}\;\mathrm{Intensity}\;\mathrm{of}\;\mathrm{nonDSB}\;\mathrm{channel}\;\mathrm{over}\;\mathrm{the}\;\mathrm{rest}\;\mathrm{of}\;\mathrm{the}\;\mathrm{nucleus}}$$
where red color indicates the fluorescence signal of the antibody against the non DSB repair protein and green color corresponds to DSB foci marker and blue is for the DAPI staining of nucleus. When $P_{\mathrm{clc}}$ tends towards zero, reversed colocalization is implied, whilst when $P_{\mathrm{clc}}$ ~1 random staining occurs, and for $P_{\mathrm{clc}}$ >1 true colocalization takes place. This parameter, like $R_{\mathrm{col}}$, conserves DSB topology and is very sensitive: It detects colocalization even under high background signal conditions. Pclc has already been incorporated to one foci software so far, namely to the JQuantPro developed by Dr. Pavel Lobachevsky [25], which is a later version of JCountPro software [117]. Its concept is quite simple, making it workable for users of almost any other software. For more details on how to incorporate $P_{\mathrm{clc}}$ into your analysis, as well as for additional applications, please refer to [25,31,116].

### 3.6. DNA Sequencing for Genome-Wide Nucleotide-Resolution DNA Damage Identification

DNA sequencing has just entered the arena of DDR detection, although it was a widely used technique in other applications. Until recently, the short-read technologies, though precise, had the ability to accurately detect only genome point-mutations like base deletions or substitutions, due to their inability to span larger DNA lengths. In case of more complex type of abnormalities, they fail to do so, due to their intrinsic limitations. The detection of more complex chromosomal aberrations (e.g., genomic rearrangements or structural variations) requires―in addition to short-read technologies―the assistance of long-read technologies, for precise characterization. This adaptation offers confidence in the detection of alterations that may span kb of DNA. Long-read sequencing technologies are already applied clinically towards the aim of personalized treatments. They recently started to attract interest in the detection of DNA damage long-term consequences, to find genomic rearrangements such as deletions, translocations, or inversions with high accuracy, low cost, and in a relatively short time. Sequencing adaptations suitable for DNA damage induction demand a site-specific DSB labeling step. Such techniques are NGR, BLISS (UMIs), BLESS, and i-BLESS:NGR detects and labels the nick-gaps, induced during DNA replication [118].BLISS can detect DSBs induced by endonucleases, using their corresponding unique molecular identifiers (UMIs) [119].BLESS is able to detect DSBs at nucleotide resolution. It is independent of proteins that bind to DNA or single-stranded DNA, which are both sources of bias. BLESS’ innovation is that it uses an amplification step for the fragments created by two DSBs [120].i-BLESS is a BLESS adaptation that is suitable for very small and fragile cell genomes (yeast), while it achieves the incredible detection accuracy of one DSB per 10^5^ cells [121].

A very recent adaptation, qDSB-Seq, combines both long- and short-sequencing techniques in order to quantify absolute numbers of DSBs. In order to quantify the DSBs of interest, spike-ins DSBs are induced to the genome, using site-specific endonucleases. The presence of spike-ins DSB offers a measure for DSB number counting calibration [122].

To conclude, DNA sequencing nowadays offers high throughput and it can substantially give added value in the analysis of the induction of DSBs and their consequences, concerning chromosomal instability. Up to now, the detection of DNA damage induction has been limited to DSB detection that can be traced in pre-known genome loci [123]. A promising field that is currently under intense investigation is the identification of the consequences of DSBs on the genome after repair mechanisms have taken place. To this end, long-read technologies (e.g., Nanopore sequencing) have developed structural variation analysis pipelines that utilize neural networks and optimized machine learning algorithms. The use of such methods can reduce the so-far existing technological limitations that led to the inability of whole genome analysis in the field of DSB-induced genomic alterations.

## 4. Biological Response to Clustered DNA Damage and Its Significance

The potential importance of clustered DNA damage is highlighted through evidence of associated biological effects, such as mutagenesis, carcinogenesis, lethality, and reduced reparability [11,124]. As discussed, strong theoretical (MC simulations) and experimental evidence suggests an increment of the complexity of DNA damage and therefore repair resistance with increasing LET [11,23,70,73,125,126,127,128,129] (Table 2). Significant improvements have been made recently towards the identification of key parameters relating to the efficient detection of complex DNA damage [25]. However, the molecular mechanisms and the basis of the underlying repair pathway choices that eventually lead to the observed biological consequences from DNA damage clustering are not yet fully understood.

### 4.1. The Role of Delayed Repair 

Complex DNA lesions including DSB and non-DSB clusters are a major challenge for the DNA repair mechanisms and the fate of a cell. The biological significance of clustered DNA lesions formed by multiple ionizations relates to the inability of cells to process them efficiently compared to isolated DNA damages, and the outcome in case of erroneous repair can vary from cell apoptosis to mutations and chromosomal instability [136]. It has been shown that clustered damaged sites (DSB and non-DSB clusters) compromise the base excision repair (BER) pathway leading to the lifetime extension of the lesions within the cluster, compared to isolated ones; thus, the chances that the lesions persist to replication and induce mutation are drastically increased. Chromosomal abnormalities can originate from the repair of DNA damage via the less accurate pathways of non-homologous (NHEJ) or alternative non-homologous end joining (alt-EJ) [137].

Complex DNA damage is known to have a slow rate of repair and to require the activation of DDR via the ATM pathway [138]. Sutherland et al. reported that the non-DSB clustered lesions generated in mammalian cells by IR show a considerably longer lifetime compared to isolated DNA lesions [21,139]. Recently, mixed beam (alpha particles and X-rays) irradiation of human peripheral blood lymphocytes revealed a lagging repair of the induced damage, presumably contributing to increased damage misrepair, particularly of complex DSBs [131]. Large 53BP1 foci formation presumably at sites of clustered DNA damage, to the detriment of small foci generated in dispersed, simple damage areas, indicates that DDR network is mainly engaged by the alpha-particles-induced complex damage, so that the low-LET radiation-induced damage is detained. Such results mechanistically demonstrate how cells cope with concurrent clustered and simple DNA damage, but also show that health effects of mixed beams may be greater than expected due to additivity of the individual beam components.

Repair of base damage within a heavily clustered damage site is likely to lead to the generation of additional DSBs, either through mistimed endonuclease action at base damage on complementary DNA strands or through interaction of base damage with the replication machinery [140]. Such DSBs generated as repair intermediates may be surrounded by unrepaired base damages and become an unrepairable, complex DSB. Complex DSBs either induced directly by IR or by the processing of non-DSB clustered lesions are repaired by slow kinetics or left unrepaired and cause cell death or pass mitosis. In surviving cells, large deletions, translocations, and chromosomal aberrations are observed [23]. An additional factor contributing to the increased mutagenicity of complex DNA damage is DNA repair deficiencies. Previous evidence suggests that deficiencies in repair enzymes like DNA-PK, APE1, and others significantly elevate the accumulation of clustered DNA lesions and genomic instability as reviewed in [141]. Last but not least, one should not disregard the environmental stress fueling DNA damage in cells and tissues through either direct induction of DNA lesions (radiation, chemicals) and epigenetic changes like DNA methylation of key repair gene promoters negatively impacting repair efficiency [142]. Whilst the delayed or incorrect repair of clustered DNA lesions in human results in the generation of mutations and genetic instability in normal tissue, an “optimistic” idea could be that it could also positively serve to cell killing of tumor cells.

### 4.2. Double-Strand Break Clustering

As already stated, damage complexity is usually determined by the existence of extra lesions in the vicinity of a DSB. Another level of complexity that can compromise lesions’ processing by chromatin destabilization in the area surrounding a cluster is DSB clustering. The formation of DSB clusters is induced by the exposure of cells to IR (particularly of high-LET radiation [143]). Studying their processing will allow us to elucidate whether the nonrandom distributions of breaks produced by high-LET particle tracks have any effect on their repair and biological effectiveness [144].

Extensive mathematical modeling provides theoretical support for the biological severity of DSB clustering [17,22,126,127,145]. However, the majority of experimental techniques used to detect IR-induced DNA damage does not provide a lot of information about the precise number and spatial arrangement of the lesions within one or two helical turns of the DNA, thus leading to an uncertainty over the biological responses. Towards this direction, revolutionary advances have been made in methods allowing the generation of enzymatic DSBs at random or in well-defined sites in the genome (by the use of *I-SceI* meganuclease), which might throw some light on several assumptions such as the exact processing of DSB clusters and the mechanisms underpinning their severity as DNA lesions [146,147,148,149]. *I-SceI* is a rare cutting homing endonuclease that is normally absent from the human and mouse genomes. *I-SceI* recognition sequences can be artificially introduced in a genome and the subsequent expression of *I-SceI* generates a DSB. To study the biological consequences of DSB clustering in a cogent manner, cell lines harboring DNA sequences, at which single DSBs and DSB clusters of known constitution are enzymatically generated, can be constructed. Efficient activation of the DDR has been demonstrated using this model system, as well as a significantly increased potential of DSB clusters, as compared to single DSBs, to kill cells and generate PARP1-dependent chromosomal translocations [87,95]. Increasing formation of chromosomal translocations is also observed at numbers and complexities, with increasing DSB-clustering. Such findings indicate that DSB repair relying on first-line DSB-processing pathways (NHEJ and HRR) is compromised within DSB clusters, presumably through the associated chromatin destabilization, consequently increasing the contribution of Alt-EJ, which has high tendency to generate chromosomal rearrangements [150]. These observations offer a mechanistic explanation for the increased efficacy of high-LET radiation. A recent review by Iliakis et al. thoroughly points to the significance of DSB clusters, introducing them as particularly consequential and underlining their effects on genomic instability and carcinogenesis [151].

### 4.3. Carcinogenesis Associated with Clustered DNA Damage

Amongst all types of DNA damage generated by IR, DSBs are considered as the most critical since they can affect cellular fate by leading to cell death or carcinogenesis when unrepaired or misrepaired [152,153,154]. One of the main etiological factors in the initiation of mutagenesis and the promotion of carcinogenesis is the lower fidelity or deficiencies of cellular repair mechanisms, which are responsible for removing or bypassing the damaged sites and restoring the initial sequence after exposure to IR. The formation of chromosomal translocations is considered as the key event that might come out of DSB processing. Translocations are well known to cause cell death and carcinogenesis [155]. Recent work also demonstrates that translocations might not be directly induced by IR, but they can be generated due to DSB processing that takes place in damaged cells by repair pathways that have evolved to process this type of lesions [153].

Considering the threats DSBs pose to cells, it comes as no surprise that the biological consequences of complex DNA damage (either DSB and non-DSB lesions, or DSB clusters) can range from point mutations and loss of genetic material to cellular lethality due to repair failure [23,135,156,157]. Singleton et al. demonstrated that irradiation-induced clustered DNA damage leads to complex genetic changes in human cells and that damage clustering is reflected in the complexity of mutations. At the sequence level, genetic changes, such as intra- and interchromosomal insertions and inversions, in association with some large deletions, were observed as a result of heavy clustering of DNA damage from α-particle irradiation [135]. Such complex genetic changes are clear indicators of genetic instability and potentially of carcinogenesis. It has also been suggested that non-DSB clusters, if unrepaired, can lead to the formation of mutations and chromosome abnormalities [158].

It is generally accepted that DSBs induced by high-LET IR are rejoined less efficiently than those induced by low-LET radiations, leaving a higher rate of non-rejoined DNA breaks [159]. High-LET radiation-induced complex DNA damage is more potently carcinogenic, generating tumors with significantly higher frequency and shorter latency compared to tumors generated by low-LET [27,160], as low LET radiation induces DNA DSBs that are rapidly repaired [23]. In terms of genomic instability, photon and particle radiation of high-LET exhibits a variety of dynamic chromosomal aberrations, including chromosomal rearrangements, such as chromosome breaks, dicentrics, translocations, and deletion mutations [161]. Through the years, scientific interest has been raised in the biological effects of IR induced DNA damage on human health and particularly the effects on astronauts during periods of space travel [162]. The potential carcinogenetic effect of space radiation was recently reviewed based on the induction of clustered DNA damage, chromosomal aberrations, micronuclei, and mutations as the main triggers of tumor initiation and progression [163].

The strong linkage between the induction of clustered DNA damage and its deficient repair, the constant DDR triggering and activation of immune system has also been considered as a possible pathway leading to carcinogenesis [31,164,165]. Maladaptive inflammation is a main pathogenic mechanism driving chronic diseases, and a key factor leading to carcinogenesis [166,167]. Chronic inflammation can result in DDR downregulation, due to the release of inflammatory mediators and reactive oxygen species (ROS) driving genomic instability. Thus, cancer susceptibility is likely to depend on specific genetic predisposition to the type and duration of this response [166]. Numerous experimental studies show inflammation to be a key process underlying the carcinogenic potential of high-LET radiation (Table 3).

## 5. Conclusions

Accumulating evidence over the years suggests that the main determinant of detrimental effects of IR is the complex DNA damage induced by this type of radiation (Figure 2). The clustered nature of DNA damage induced especially by high-LET radiations, as also suggested from the early and recent MC simulations as along with the experimental evidence, triggers biological responses differing from everyday oxidative stress [177]. In this review, we present, as inclusively as possible within the natural constraints of a paper, the advancements and challenges in the detection, simulations, and proper estimation of the biological significance of IR-induced clustered DNA damage. We currently consider the detection challenges as the most serious obstacle in establishing definite associations between IR-induced damage and prediction of biological responses not only at the cellular level, but also at the organism level.

At the same time, we provide evidence of the necessity to investigate radiation effects at a systems biology level. In Table 2 and Table 3, we have included specific examples of such different and ‘special’ biological responses due to the induction of highly complex DNA damage. Undoubtedly, from all biological responses and outcomes, genomic instability and chronic inflammation fueled by constant DDR induction and deficient or incomplete repair of damages are the key factors in the understanding and predicting the full range of radiation biological effects. Therefore, we emphasize the need for studies involving also systems biology approaches like bioinformatics and omics in general.

## Figures and Tables

**Figure 1 cancers-11-01789-f001:**
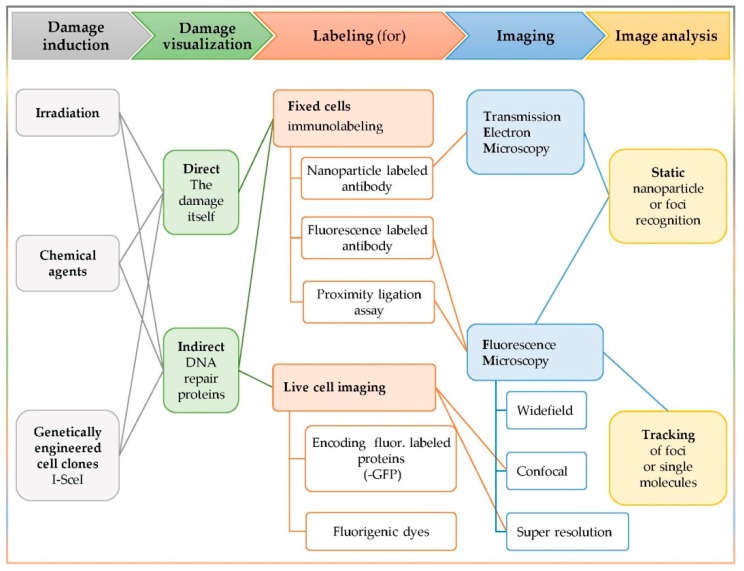
Synopsis of the common steps among clustered DNA damage in situ detection techniques. Synergistic action of involved enzymes is confirmed via detection of colocalizing signal, produced by the labeled proteins of interest.

**Figure 2 cancers-11-01789-f002:**
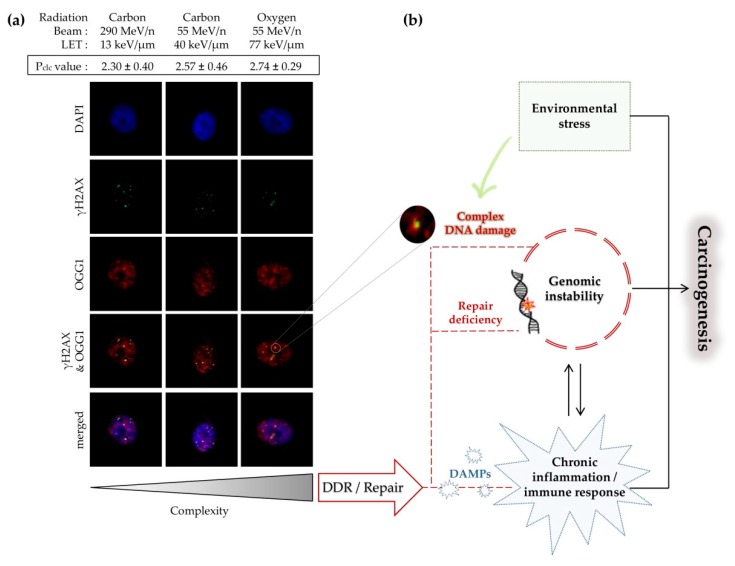
Linking processing of clustered DNA damage and its biological consequences. (**a**) Formation of fluorescent γH2AX–OGG1 clusters in normal human bronchial epithelial cells at 1 h post-irradiation with heavy ions of varying LETs (adapted from [178]). Colocalization parameter (Pclc) values (as described in Section 3.5.2) increase with increasing LET; (**b**) induction of complex DNA damage and biological consequences in mammalian cells. Long-term consequences of DNA damage include different forms of genomic instability, which significantly contribute to carcinogenesis. Furthermore, processing of unrepaired or persistent clustered DNA damage promotes cell death. Activation of DNA damage response (DDR) and repair machinery can trigger the extracellular release of diverse signatures of ‘Danger’ signals or damage-associated molecular patterns (DAMPs: ATP, short DNAs/RNAs, ROS, heat shock proteins (HSPs) and others) [179].

**Table 1 cancers-11-01789-t001:** Most widely used radiation transport and track-structure Monte Carlo (MC) codes for the simulation of radiation-induced damage.

Code	Particles	Pros	Cons	Ref.
EGS4	photons & e-	Developed specifically for dose calculations in RT applications	Not applicable at the nanoscale due to limitations of the physics models	[47]
FLUKA	Large set of particles	Multipurpose code covering medical, space, nuclear, and high-energy applications	Closed-source code. Not applicable at the nanoscale due to limitations of the physics models	[48,49]
Geant4	large set of particles	Open-source toolkit covering medical, space, nuclear, and high-energy applications. Contains powerful visualization packages. Includes low-energy physics models for sub-keV transport. Maintained by a large international collaboration	Complex toolkit. Computationally intensive. Requires users with advanced programming skills.	[41,50]
Geant4-DNA	Large set of particles	Performs event-by-event simulation for track-structure applications in liquid water. Includes a variety of low-energy electron models. Includes chemistry (water radiolysis) and biology (DNA damage and repair models) Maintained by a large international collaboration and continuously upgraded	Complex toolkit. Computationally intensive. Requires users with advanced programming skills.	[51,52]
KURBUC	Photons, neutrons, electrons, protons, alpha, carbon ions	Performs event-by-event simulations down to very low energies for track-structure applications. Includes physics models for both gaseous and liquid water medium. Includes chemistry (water radiolysis) and biology (DNA damage and repair models)	Proprietary code. Specific to water medium.	[53,54]
MC4	e-, protons, α-particles	Performs event-by-event simulations down to low-energies for track-structure applications. Includes physics models for both gaseous and liquid water medium. Known for its upgraded models for the liquid phase.	Proprietary code. Specific to water. Does not extend to relativistic energies.	[55,56,57,58]
MCDS	e-, p, alpha particles & ions	Simulates in a very fast way (ranging from seconds to minutes) the induction and clustering of DNA lesions.	Lacks accuracy. Impossible to generate damage configurations for e- with energy lower than 80 eV.	[59]
MCNP	Large set of particles	Multipurpose code covering nuclear and medical applications. Known for its accurate neutron models.	Not a free access code.	[60]
PARTRAC	e- & ions	Performs event-by-event simulation for track-structure applications. Uses physics models specifically developed for liquid water. Includes chemistry (water radiolysis) and biology (DNA damage and repair models)	Proprietary code. Specific to water. Does not include relativistic energies. The parameterization of the DNA model does not consider the DNA atomic composition.	[17,61]
PENELOPE	Photons, e- & e+	Developed for dose calculations in radiotherapy applications. Known for its electron models that extend to low-energies. Event-by-event simulations possible for application to microdosimetry	Limited track-structure applications due the incomplete simulation of electron track ends. Requires users with advanced programming skills to develop their own applications.	[62]
PEREGRINE	Large set of particles	Developed for radiotherapy treatment planning.	Gives results through a computer cluster. Not applicable at the nanoscale due to limitations of the physics models	[63]
RITRACKS	e- & ions	Performs event-by-event simulation up to relativistic energies. Simulation of radiation tracks without the need of extensive knowledge of computer programming. Simple in use.	Distributed only to authorized users. Specific to water. Uses atomic cross sections that are not reliable for nanoscale transport in liquid water.	[64]
Topas-nBio	large set of particles	Uses Geant4-DNA as its transport engine. Simply to use software specifically developed for radiobiological applications at the (sub) cellular level.	Not (yet) open-sourced. Specific to water medium.	[65]
TRAX	e- & ions	Performs event-by-event simulation for track-structure applications in various media.	Uses atomic cross sections that are not reliable for nanoscale transport in condensed-phase media.	[66]

**Table 2 cancers-11-01789-t002:** Summary of in vitro studies on biological responses to clustered DNA damage induction.

Cell Type	Radiation Type	Biological Response	Ref.
HeLa and oropharyngeal squamous cell carcinoma (UMSCC74A and UMSCC6) cells	High-LET α-particles (121 keV/μm) or protons (12 keV/μm), versus low-LET protons (1 keV/μm) and X-rays	Τhe signaling and repair of complex DNA damage, particularly induced by high-LET IR is coordinated through the specific induction of H2B_ub_ catalyzed by MSL2 and RNF20/40, a mechanism that contributes to decreased cell survival after irradiation.	[125,130]
Human peripheral blood lymphocytes	Mixed beam of alpha-particles (^241^Am source, 0.223 Gy/min, LET: 90.9 keV/μm) and X-rays (190 kV, 4.0 mA, 0-2 Gy)	Induced DNA damage was above the level predicted by assuming additivity. The activation levels of DDR proteins and mRNA levels of the studied genes were highest in cells exposed to mixed beams. The repair of damage occurs with a delay.	[131]
Human dermal fibroblasts	High-LET IR with carbon ions (9.5 MeV/n; LET 190 keV/μm; calculated mean dose: 1.52 Gy) or calcium ions (7.7 MeV/n; LET 1800 keV/μm; calculated mean dose: 14.4 Gy), versus low-LET IR with 6-MV photons (10 Gy)	High-LET-IR induced clustered DNA damage and triggered profound changes in chromatin structure along particle trajectories. DSBs exhibited delayed repair despite cooperative activity of 53BP1, pATM, pKap-1.	[26]
Normal human skin fibroblasts	^60^Co γ-rays (LET=0.3 keV/μm), accelerated ^11^B (E = 8.1 MeV/nucleon, LET=138 keV/μm) ions	It has been found that heavy charged particles induce clustered DNA damage in the genome of cells that can lead not only to gene mutations, but also to large deletions.	[132]
HeLa Kyoto cells	Pulsed UV laser (micro-irradiation, IR)	Recruitment and dissociation of 70 DNA repair proteins to laser-induced complex DNA lesions.	[83]
Human uveal melanoma (92–1) cells	Carbon ions (LET: 80 keV/μm) and iron ions (LET: 400 keV/μm) at different doses, versus X-rays (LET: 4 keV/μm)	Heavy ions were more effective at inducing senescence than X-rays. Less-efficient repair was observed when DNA damage was induced by heavy ions compared to X-rays and most of the irreparable damage was complex of SSBs and DSBs, while DNA damage induced by X-rays was mostly repaired in 24 h. Results suggest that DNA damage induced by heavy ion is complex and difficult to repair, thus presenting as persistent DNA damage, and pushes the cell into senescence.	[128]
Human dermal fibroblasts, NFFh-TERT foreskin fibroblasts	Low-LET irradiation with 6 MV photons, versus high-LET irradiation with carbon ions (9.5 MeV/n; LET = 190 keV/μm)	High-LET irradiation caused localized energy deposition within the particle tracks and generated highly clustered DNA lesions with multiple DSBs in close proximity. Ηuge DSB clusters predominantly localized in condensed heterochromatin. High-LET irradiation-induced clearly higher DSB yields than low-LET irradiation, and large fractions of these heterochromatic DSBs remained unrepaired.	[24]
Human osteosarcoma cell line (U2-OS)	X-rays (250 keV, 16 mA; LET: 2 keV/μm), versus heavy ions: ^238^U ions (LET: 15,000 keV/μm), ^207^Pb ions (LET: 13,500 keV/μm), ^197^Au ions (LET: 13,000 keV/μm), ^119^Sn ions (LET: 7,880 keV/μm), ^59^Ni (LET: 3,430 keV/μm), ^48^Ti (LET: 2,180 keV/μm), ^14^N ions (LET: 400 keV/μm), and ^12^C ions (LET: 170 keV/μm)	DSB complexity plays a critical role in the decision for DSB end-resection in G1-cells. CtIP, MRE11, and EXO1 are required for the resection of complex DSBs in G. Repair of complex DSBs relies on resection independent of the cell cycle stage.	[133]
Human cells (fibroblasts, HBECs)	1 Gy of Si (LET: 44 keV/μm) or Fe (LET: 150 keV/μm) ions	Direct visualization of clustered DNA lesions at the single-cell level using 53BP1, XRCC1, and hOGG1 as surrogate markers for DSBs, SSBs, and base damage, respectively, reveals that most complex DNA damage is not repaired in human cells. Unrepaired clustered DNA lesions result in the generation of a spectrum of chromosome aberrations. Checkpoint release before the completion of clustered DNA damage repair is a major cause of chromosomal aberrations.	[23]
Human Lung Adenocarcinoma (A5490) cells	^12^C ions (62 MeV, LET: 290 keV/μm), versus ^60^Co γ-rays (1–3 Gy)	Carbon ions were three times more cytotoxic than γ-rays. The observed decrease in number of γ-H2AX foci 4 h after γ-rays irradiation indicates repair of damage and is supported by nearly 100% survival, whilst the decrease in γ-H2AX foci after carbon ion irradiation was not indicative of repair.	[134]
HF12 primary male human fibroblast cells	^238^Pu α-particles (range, ∼20 μm; peak energy, 3.26 MeV; LET=121.4 keV/μm)	Many α-particle-induced mutations are large deletions. Rejoining at microhomologies characterizes large deletion junctions. Intra- and interchromosomal insertions and inversions occur at the sites of some large deletions. Novel fragments found in complex rearrangements derive from other sites of radiation damage in the same cell.	[135]

**Table 3 cancers-11-01789-t003:** High-linear energy transfer (LET) radiation triggering immune responses.

Biological System/Cell Type	Radiation Type	Immune Response	Ref.
Peripheral blood mononuclear cells (PBMCs) of head and neck (HNSCC) cancer patients	Intensity modulated radiotherapy (IMRT), (51–74 Gy total dose, 1.6–3 Gy dose/fraction)	Expression of the FXDR, SESN1, GADD45, DDB2, and MDM2 radiation-response genes were altered in the PBMCs of patients after RT. All changes were long-lasting, detectable one month after RT. Local tumor irradiation induces systemic changes in the level of immune and inflammation-related plasma proteins. RT induces changes in the immune phenotype of PBMCs of HNSCC patients.	[168]
Murine CT26 colorectal cancer cells	8 Gy proton beams at 1.09 keV/μm (low), 2.58 keV/μm (medium) and 7.7 keV/μm (high) LET.	Increase in percentage expression of immune markers (OX40L, CD40, ICAM-1, and MHC-I) in high-LET irradiated cells. High-LET proton radiation can be used to stimulate better immunogenic phenotype in tumor cells compared to low LET proton radiation.	[169]
Human cancer cell lines: TE2, KYSE70, A549, NCI-H460 and WiDr	Carbon ions (290 MeV/n, LET 30 keV/µm)	Carbon-ion beams significantly increased HMGB1 (a damage-associated molecular pattern—DAMP) levels in the culture supernatants.	[170]
NCI-H446 (lung tumor cells)	Carbon ions (290 MeV/n, LET 13 keV/µm)	Cyto- and chemokine response release by tumor cells after irradiation. (TNF-α)	[171]
Tumor-bearing mice (C3H/He, Balb/c nude mice)	Carbon ion irradiation (290 MeV/n, LET=77 keV/µm)	Increased cytotoxic T-lymphocytes (CTL)-associated lysis of isolated tumor splenocytes after carbon ion irradiation treatment with supplementary intratumoral dendritic cell (DC) injection.	[172]
Tumors of mouse squamous cell carcinoma (NR-S1) cells inoculated in the legs of C3H/HeSlc mice	Carbon ions (290 MeV/n, 6-cm spread-out Bragg peak, 6 Gy)	Even when exposed to the same equivalent doses, carbon ion therapy might activate the immune system to a greater extent than conventional RT.	[173]
NR-S1 and SCCVII (squamous cell carcinoma), NFSa, #8520 (fibrosarcoma)	Carbon ions (290 MeV/n, LET 50 keV/µm)	Significant C-ion induced upregulation of stress-responsive and cell-communication genes common to different tumor types.	[174]
Rat skin	^56^Fe ions (1.01 GeV/n)	^56^Fe ion radiation significantly induced inflammation-related genes, including many in the categories of ‘immune response’, ‘response to stress’, ‘signal transduction’, and ‘response to biotic stress’, that contribute to carcinogenesis.	[175]
Highly aggressive HT1080 human fibrosarcoma and LM8 mouse osteosarcoma cells	Carbon ion beams (290 MeV/n), versus X-rays	When compared with photon irradiation, carbon ion exposure reduced the number of distant lung metastasis in carcinoma models in immunocompetent mice.	[176]

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
