# Peer review of "Ionizing Radiation and Complex DNA Damage: From Prediction to Detection Challenges and Biological Significance"

_cancers, 2019, doi:10.3390/cancers11111789_

Round 1
Reviewer 1 Report
In this review by Mavragani et al, the often overlooked concept of Complex DNA damages are discussed, on the level of prediction, detection and biological significance.
The goal for the review is set high which is excellent, however to some degree I think that this ambition has resulted in the review becoming to difficult for someone lacking the physics and mathematical knowledge and to shallow, with details lacking, for someone who is more familiar with the field.
In addition the language throughout the text make it even more difficult to grasp the content without knowing beforehand what is meant. This is often due to unusual choices of prepositions and complicated sentence constructions.
I also have a few more specific suggestions;
On page 4: here a very thorough review of Codes used for analysis of MC simulations is presented. It would be more transparent, and easier to get an overview of pros and cons of the same, if they were presented in a Table.
When it comes to the listing of DNA damage detection methods I believe that today sequencing based procedures also need to be presented, such as "BLESS", BLISS" and iBLESS and similar, either one by one or as a concept (for a review Bouman and Crosetto 2018).
Author Response
Response to reviewers’ comments
We sincerely thank all the reviewers for their positive comments and for all their efforts to improve our manuscript. We also have to acknowledge the fast and efficient handling of our manuscript by Ms. Elena Ma, Assistant Editor, MDPI AG. We revised the manuscript accordingly and we corrected some grammatical errors and the text in order to secure better understanding. We now hope that this revised version of our manuscript stands better and we have better communicated our message. Please find below the analytical response to the reviewers’ comments:
REVIEWER REPORT(S):
Reviewer #1Comments and Suggestions for Authors
In this review by Mavragani et al, the often overlooked concept of Complex DNA damages are discussed, on the level of prediction, detection and biological significance. The goal for the review is set high which is excellent, however to some degree I think that this ambition has resulted in the review becoming to difficult for someone lacking the physics and mathematical knowledge and to shallow, with details lacking, for someone who is more familiar with the field. In addition the language throughout the text make it even more difficult to grasp the content without knowing beforehand what is meant. This is often due to unusual choices of prepositions and complicated sentence constructions. I also have a few more specific suggestions; On page 4: here a very thorough review of Codes used for analysis of MC simulations is presented. It would be more transparent, and easier to get an overview of pros and cons of the same, if they were presented in a Table. When it comes to the listing of DNA damage detection methods I believe that today sequencing based procedures also need to be presented, such as "BLESS", BLISS" and iBLESS and similar, either one by one or as a concept (for a review Bouman and Crosetto 2018).
Response: We really welcome the reviewer’s comments. Please find below the analytical responses to the reviewers’ suggestions:
“…review becoming too difficult for someone lacking the physics and mathematical knowledge and too shallow, with details lacking, for someone who is more familiar with the field.”
R.: We tried to address this issue by providing more technical information, especially in sections describing microscopy techniques and image analysis.
”…unusual choices of prepositions and complicated sentence constructions.”
R.: We revised the manuscript accordingly and we corrected some sentences in order to secure better understanding.
“It would be more transparent, and easier to get an overview of pros and cons of the same, if they were presented in a Table”
R.: Concerning the part on the Monte Carlo codes, we address the issue raised by the reviewer and we added a new Table (1) which contains a list of the most widely used codes, along with an overview of their main pros and cons.
“When it comes to the listing of DNA damage detection methods I believe that today sequencing based procedures also need to be presented, such as "BLESS", BLISS" and iBLESS and similar, either one by one or as a concept (for a review Bouman and Crosetto 2018).”
R.: A paragraph (3.6) is added where DNA sequencing for genome-wide nucleotide-resolution DNA damage identification is presented. The techniques proposed by the reviewer were also discussed.
Reviewer 2 Report
This is a very well written article that provides an excellent overview of radiation induced clustered DNA damage. The article discusses the complexity of the nature of DNA lesions and their prediction through modeling and simulation, techniques to detect these in cells, and the DNA damage response to such lesions. Overall the article will serve as a great resource to the research community in the field of DNA damage and radiation.
Author Response
Response to reviewers’ comments
We sincerely thank all the reviewers for their positive comments and for all their efforts to improve our manuscript. We also have to acknowledge the fast and efficient handling of our manuscript by Ms. Elena Ma, Assistant Editor, MDPI AG. We revised the manuscript accordingly and we corrected some grammatical errors and the text in order to secure better understanding. We now hope that this revised version of our manuscript stands better and we have better communicated our message. Please find below the analytical response to the reviewers’ comments:
Reviewer #2
Comments and Suggestions for Authors
This is a very well written article that provides an excellent overview of radiation induced clustered DNA damage. The article discusses the complexity of the nature of DNA lesions and their prediction through modeling and simulation, techniques to detect these in cells, and the DNA damage response to such lesions. Overall the article will serve as a great resource to the research community in the field of DNA damage and radiation.
R: We really appreciate the reviewer’s positive feedback.
Reviewer 3 Report
This is a useful review on radiation-induced DNA complex damage, covering both theoretical and experimental aspects and suggesting that, in the future, emphasis should be given to the links between the induction of complex damage, its processing and systemic effects like genomic instability and immune responses.
The paper is well organized and rather well written, and I only suggest the (minor) changes below:
line 38: physics and biology characteristics -> physical and biological characteristics
line 40: as a reference n. 3, instead of Ballarini et al Radiation Research 2013 I would quote this more recent work by the same group of authors: MP Carante, C Aimè, JJ Tello Cajiao and F Ballarini, BIANCA, a biophysical model of cell survival and chromosome damage by protons, C-ions and He-ions at energies and doses used in hadrontherapy. Phys Med Biol 63(7),075007, 2018.
l 41: focuses also to -> focuses also on
l 48: ionizations' radius -> ionization radius
l 49: per volume unit -> per unit volume
l 57: it is increasingly.. -> it has increasingly...
l 70: in clinic -> in clinics
l 79: invented -> developed
l 80: relative high -> relatively high
line 85: besides LEM and MKM, here it may be worth mentioning that the BIANCA model, in addition to cell survival [see reference above], also predicts chromosome aberrations, which can be regarded as indicators of normal tissue damage: Tello et al 2018, Proximity effects in chromosome aberration induction: dependence on radiation quality, cell type and dose. DNA Repair 64, 45-52.
line 192: a molecular... -> of a molecular...
l 199: allow -> allows
l 242: there has been proposed -> it has been proposed
l 253: is consisted of... -> consists of...
l 255: DNA damage DNA -> DNA damage
l 390: it could be also be used -> it could be also used
l 453: serves -> serve
l 462: here the authors may consider to also quote the following work on DSB clustering: MP Carante et al 2015, Modelling radiation-induced cell death: role of different levels of DNA damage clustering. Radiat Environ Biophys 54(3), p 305-316.
l 482: I would change the title as follows: Carcinogenesis associated to clustered DNA damage
l 541: they key factors -> the key factors
Author Response
Response to reviewers’ comments
We sincerely thank all the reviewers for their positive comments and for all their efforts to improve our manuscript. We also have to acknowledge the fast and efficient handling of our manuscript by Ms. Elena Ma, Assistant Editor, MDPI AG. We revised the manuscript accordingly and we corrected some grammatical errors and the text in order to secure better understanding. We now hope that this revised version of our manuscript stands better and we have better communicated our message. Please find below the analytical response to the reviewers’ comments:
Reviewer #3Comments and Suggestions for Authors
This is a useful review on radiation-induced DNA complex damage, covering both theoretical and experimental aspects and suggesting that, in the future, emphasis should be given to the links between the induction of complex damage, its processing and systemic effects like genomic instability and immune responses. The paper is well organized and rather well written, and I only suggest the (minor) changes below:
R: We really appreciate the reviewer’s constructive and useful comments which we have attended carefully all. In addition, all suggested references on the BIANCA model and other papers proposed have been added and discussed accordingly.
As seen below:
line 38: physics and biology characteristics -> physical and biological characteristics
line 40: as a reference n. 3, instead of Ballarini et al Radiation Research 2013 I would quote this more recent work by the same group of authors: MP Carante, C Aimè, JJ Tello Cajiao and F Ballarini, BIANCA, a biophysical model of cell survival and chromosome damage by protons, C-ions and He-ions at energies and doses used in hadrontherapy. Phys Med Biol 63(7),075007, 2018.
l 41: focuses also to -> focuses also on
l 48: ionizations' radius -> ionization radius
l 49: per volume unit -> per unit volume
l 57: it is increasingly.. -> it has increasingly...
l 70: in clinic -> in clinics
l 79: invented -> developed
(ü) l 80: relative high -> relatively high
line 85: besides LEM and MKM, here it may be worth mentioning that the BIANCA model, in addition to cell survival [see reference above], also predicts chromosome aberrations, which can be regarded as indicators of normal tissue damage: Tello et al 2018, Proximity effects in chromosome aberration induction: dependence on radiation quality, cell type and dose. DNA Repair 64, 45-52.
line 192: a molecular... -> of a molecular...
l 199: allow -> allows
l 242: there has been proposed -> it has been proposed
l 253: is consisted of... -> consists of...
l 255: DNA damage DNA -> DNA damage
l 390: it could be also be used -> it could be also used
l 453: serves -> serve
l 462: here the authors may consider to also quote the following work on DSB clustering: MP Carante et al 2015, Modelling radiation-induced cell death: role of different levels of DNA damage clustering. Radiat Environ Biophys 54(3), p 305-316.
l 482: I would change the title as follows: Carcinogenesis associated to clustered DNA damage
l 541: they key factors -> the key factors